# PBPK Evaluation of Sofosbuvir Dose in Pediatrics Using Simcyp®

**Rania Elkeeb [1], Anomeh Avartoomian [1], Amira S. Gouda [2], Ahmed M. Abdel-Megied [3], Ola Abdallah [4] and Eman Atef [1,*]**

1   School of Pharmacy, West Coast University, Center of Graduate Studies, Los Angeles, CA 90004, USA
2   Bioequivalence Research Department, Zi Diligence Bio-Equivalence Center, Cairo 11571, Egypt
3   Department of Pharmaceutical Sciences, Notre Dame of Maryland University, School of Pharmacy, Baltimore, MD 21210, USA
4   Analytical Chemistry Department, Faculty of Pharmacy (For Women) Al-Azhar University, Cairo 11765, Egypt
*   Correspondence: eatef@westcoastuniversity.edu

**Abstract:** The aim of the study is to evaluate the effectiveness of the pediatric sofosbuvir weight-based dosing strategy in providing an equitable drug exposure compared to the marketed dose. The physiologically based pharmacokinetic (PBPK) modeling and simulation is a valuable tool in assessing drug dosing and toxicity in populations with physiological, pathological, and genetic pharmacokinetic (PK) variability. The PBPK model of the sofosbuvir compound was developed using Simcyp® V20. The model was developed and verified using the published sofosbuvir's physicochemical properties and clinical data from multiple studies on healthy adult volunteers, hepatitis C virus (HCV)-infected adults, and HCV-infected pediatrics. The AUC and $C_{max}$ fold ratio of (predicted/observed) fell within the acceptable range of 0.5–2 in all tested adults' data, confirming the successful development of the sofosbuvir Simcyp® compound model. Using this model, a weight-based dosing regimen of 6 mg/kg in pediatric patients was simulated and compared to the 150 mg and 200 mg approved dose for 3–6 and 6–12 y/o pediatric patients, respectively. No dose adjustment was recommended in patients ages 6–12 y/o. However, compared to the approved 150 mg for 3–6 y/o, the weight base dose provided an equitable drug exposure to adults. Further clinical studies are warranted to verify this finding.

**Keywords:** sofosbuvir; PBPK; pediatrics; Simcyp®; HCV; dose adjustment; dose simulation

## 1. Introduction

Hepatitis C virus (HCV) is a global health burden [1]. It is a major cause of liver cirrhosis and hepatocellular carcinoma, and it may lead to liver failure and death. The HCV affects both adults and children. Approximately 150 million people worldwide have chronic HCV infection, with at least three million additional people becoming infected every year. Vertical transmission during gestation or delivery is one of the main modes of infection. The estimated rate of chronic infection is more than 3.4 million viremic children worldwide [2]. As the number of HCV infections in childbearing age women has increased, this surely suggests that the number of infants born with HCV infections will also increase [3].

The acute HCV infection can be associated with a few leading symptoms, such as jaundice, and it can result in high blood alanine aminotransferase levels. On the other hand, the infection is often asymptomatic, and 20% of infected patients are unaware of it. This suggests the need for enhanced identification of HCV-infected pregnant women and adding HCV as routine screening during pregnancy [3]. Thus, treating children is vital in avoiding the development of liver disease [2,4].

Before the discovery of directly acting antivirals (DAA), the treatment of chronic HCV was based completely on a combination of pegylated interferon and ribavirin. Fortunately, with the discovery of the DAA in 2014, the chronic HCV infection became curable in almost

100%. The DAA regimens resulted in a paradigm shift in the handling of the disease. This is one example of victory in the fight against infections. It allowed for the complete global eradication of the virus [5]. The high rates of HCV replication and possible mutation permit HCV to rapidly mutate. Using the combination DAA therapy in targeting different viral functions and stages of the virus life cycle is an approach to limit resistance. A similar tactic is applied when using cocktail anti-HIV to avoid resistance to monotherapy with the difference that the DAA therapy is administered for a limited time, unlike the HIV therapy, which is a lifetime therapy [5].

Sofosbuvir is a DAA of distinct interest due to its effectiveness, low side effects, oral administration, and high barrier to resistance.

Sofosbuvir is a potent oral nucleotide analogue prodrug. It is a DAA agent used against HCV as part of a combination therapy. Sofosbuvir's active metabolite blocks the NS5B polymerase, an essential target protein for HCV replication, and it is found to be a good target for DAAs [6]. A sofosbuvir combination with ledipasvir has been used for adults for several years.

Sofosbuvir is well absorbed. At least 80% of a sofosbuvir dose is recovered in the urine as the parent drug and its metabolite. Being a p-glycoprotein (P-gp) substrate, sofosbuvir absorption is increased by P-gp inhibitors. The drug has moderate protein binding, at approximately 82% [7].

The sofosbuvir site of action is the hepatocytes, where it is activated to GS-331007 triphosphate metabolite. This metabolite is a highly soluble ionized molecule; thus, it is impermeable through the cell membrane and undetectable in the plasma.

The GS-331007 triphosphate is eventually inactivated to the GS-331007 metabolite, which is renally eliminated. The GS-331007 is the analyte used in clinical studies as it accurately accounts for systemic exposure. Sofosbuvir and its metabolites are not inhibitors of any human CYP isozymes [8].

Motivated by the universal acceptance that in silico calculations and in vitro data can provide insight into pharmacokinetic processes and reduce the time, effort, and money spent in the drug development process, many pharmaceutical and biotechnology companies have heavily applied modeling and simulation in pharmacokinetics. With many of the regulatory institutions such as the FDA recognizing the in silico supporting data and platforms, PBPK models are now used to supplement clinical trials [9].

It has been broadly applied to predict clinical PK outcomes in the presence of patient-related factors such as age, genetic factors, organ impairment, and pregnancy, or administration-related factors such as formulation, frequency, food, or co-administered drugs. This is found to be especially helpful for rare diseases or pediatric clinical studies and other studies with limited subjects and patient populations. It is a challenge, on the other hand, in the development of these virtual populations due to the limitation of available data and enrollment challenges or ethical issues [10].

Many current in silico user-friendly tools allow flexibility to design models of concepts that go beyond the software capabilities, which allows for the broader utilization of the software for many scientists who have no programming proficiencies [11]. A few examples of simulation software programs used for this purpose are Simcyp® Gastro Plus™, GI-Sim, and PK Sim. Some PK research groups prefer to develop their own in-house models should they have the needed coding skills.

A few studies have aimed to explore the strengths and restrictions of simulation software programs. The average performance is relatively consistent across many software platforms, and the accuracy of the prediction is dependent on the physicochemical properties of the drug and the application or goal of the study.

Some programs have better prediction for drugs with challenging solubility and incomplete absorption, while others create more accurate models for drugs with challenging elimination and metabolism or models involving drug–drug interactions. Some studies suggest that modeling and simulation groups should perform model evaluations using

multiple software to decide on the best model, but the reality is that there is a major cost hindrance to this ideal practice [9,12–14].

Our study used the Simcyp® Simulator, a PBPK modeling and simulation software that aids in mechanistic and quantitative prediction of the impact of diseases, age, and different race genomic composition.

Simcyp® V20 software is composed of a whole-body physiologically based pharmacokinetic (PBPK) model that integrates drug physicochemical and physiological information with clinical trial data to make PK predictions in different populations. The simulator is especially imperative in populations involving limited subjects for ethical and legal reasons, such as pediatrics [15].

The adult dose of sofosbuvir is 400 mg/day for several weeks, depending on the virus genotype. Later, the FDA-approved pediatric dose of sofosbuvir for children ≥ 6 y/o weighing ≥ 17 kg is 200 mg tablet or pellets once daily. For patients < 6 y/o or weighing < 17 kg, the dose is 150 mg pellets once daily [15]. The details of the currently approved FDA doses are summarized in Table 1. The aim of this study is to investigate the impact of age on the sofosbuvir pediatric dose and to assess the currently approved dose. The modeling, verification, and simulation approach is used to quantify the age effect on sofosbuvir PK by integrating the adult clinical PK data with the pediatric physiological information to make a recommendation on an optimal pediatric dose.

**Table 1.** Approved FDA adult and pediatric doses.

| Population | Dose |
|---|---|
| Healthy Adults | 400 mg |
| Pediatric: Currently Approved | |
| 3–6 yo < 17 kg | 150 mg |
| 6–12 yo | 200 mg |

## 2. Methods

The overall study workflow is summarized in Figure 1.

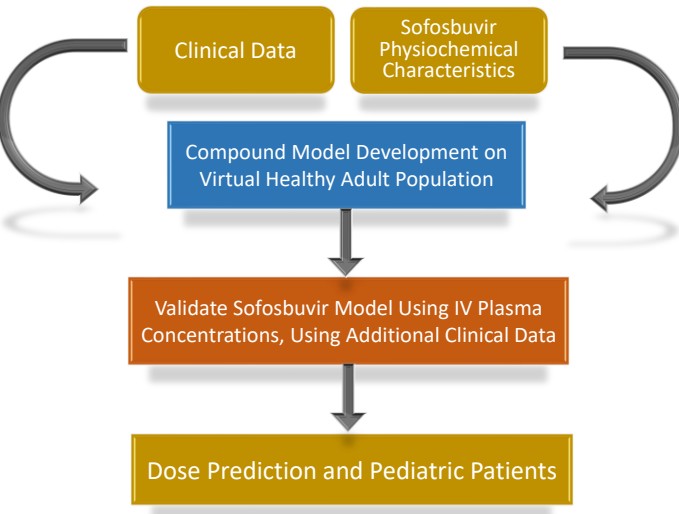

**Figure 1.** Flowchart of the study design.

The sofosbuvir compound model is not among the developed molecules in the Simcyp® compounds library nor shared by any member of the Simcyp® community. Thus, the Simcyp® software (Version 20, Simcyp Division, Certara UK Limited, Sheffield, UK) [16] was used in developing a new sofosbuvir compound model in the healthy adult population using the published clinical data with an inclusion/exclusion criterion.

Afterward, the model was validated using different sets of published clinical data. Subsequently, a literature search for sofosbuvir clinical PK data in the pediatric population was carried out to validate the sofosbuvir model in pediatrics [17] and to use it to predict a pediatric dose and compare it to the approved FDA dose.

Plot digitizer software is a tool used to reverse-engineer plot images to extract underlying numerical data. In this study, the sofosbuvir PK parameters and the plasma profiles were plot-digitized using Web Plot Digitizer version 4.4. [18]. The feathering method was used to calculate the $k_e$, $k_a$, and AUC parameters to confirm the reported data.

## 2.1. Model Development

The approach for developing the PBPK model followed the best practices including development, optimization, and verification in the adult population [19]. The sofosbuvir published physicochemical and PK parameters were used to build a PBPK model in Simcyp® V20 using the virtual healthy adult population as listed in Table 2 [4].

**Table 2.** Pharmacokinetic parameters used in building sofosbuvir Simcyp® model.

| Physicochemical Parameter | Value | References |
|---|---|---|
| Molecular Weight (g/mol) | 529.45 | |
| Log P | 1.6 | [20] |
| Compound Type | Weak Acid | [20] |
| Pka | 9.3 | [20] |
| B/P | 0.71 | [20] |
| Fu | 0.17 for healthy adults | [20] |
| Absorption Model: Advanced Dissolution Absorption and Metabolism (ADAM) | | |
| GI Peff ($10^{-4}$ cm/s) | 7 | Predicted based on sensitivity analysis |
| Formulation | Solid formulation Immediate release Dissolution Profile | Dissolution profile in supplemental data |
| Distribution Model: Full PBPK Model | | |
| Vss (L/kg) | 1.97 | Predicted Simcyp® method 1 |
| Elimination: Enzyme Kinetics | | |
| $Cl_{int}$: Recombinant CES1 (μg/mL/mg) | 49 | Sensitivity analysis |
| CLR (L/h) | 14.2 | [7] |

B:P, blood to plasma ratio; fu, fraction unbound in plasma; logP, log octanol–water partition coefficient for neutral compound; MW, molecular weight; pKa, acid dissociation constant; GIT Peff, effective GIT permeability; Vss, volume of distribution at steady state; CLint, intrinsic clearance; CLR, renal clearance.

The compound model was developed using the Distribution Full PBPK model, the ADAM Absorption model, and the Enzymatic Clearance.

The sensitivity analysis was performed to predict the clearance as the intrinsic clearance ($Cl_{int}$), the maximum reaction rate ($V_{max}$), and the Michaelis Constant (km) of the carboxylesterase; the main metabolizing enzymes are not published.

Even though our data search revealed multiple published clinical studies on sofosbuvir, the Hill et al. study was the one used to develop the model since it included the highest number of participants [21]. The $C_{max}$, $t_{max}$, and AUC were calculated after plot-digitizing the data using Web Plot Digitizer version 4.4.

### 2.2. Model Verification

The simulated PK parameters were compared with the observed values of additional sofosbuvir adult clinical studies for verification purposes [22–24]. In this analysis, healthy adult data were included and data from subjects with cirrhosis and pregnant women were excluded for the potential difference in extraction ratio and enzymatic activities [22,25]. In addition, clinical studies with high intersubject variability in both $C_{max}$ and AUC were excluded [20,26–29].

The simulated values were used to calculate the (predicted/observed) fold ratios, a frequently used criterion to verify the compound model.

### 2.3. Paediatric Simulation

The PK parameters were simulated in the Simcyp® virtual pediatrics population using the developed sofosbuvir model. The simulated data were compared to pediatric clinical data (Table 3).

**Table 3.** Comparison of physiologically based pharmacokinetic (PBPK) model prediction and clinically observed values for PK parameters of sofosbuvir in pediatrics population.

| Observed Data (Age Group) | AUC (µg·h/L) | | | $C_{max}$ (µg/L) | | |
|---|---|---|---|---|---|---|
| | Pred | Obs | Fold (Pred/Obs) | Pred | Obs | Fold (Pred/Obs) |
| Schwarz et al. 150 mg (3–6 y/o) [30] | 2908 | 2500 | 1.2 | 1596 | 1420 | 1.1 |
| Rosenthal et al. 200 mg (6–12 y/o) [31] | 1842 | 960 | 1.9 | 1048 | 609 | 1.7 |
| Murray et al. 200 mg (6–12 y/o) [32] | 1842 | 1600 | 1.2 | 1048 | 906 | 1.2 |
| Simulated dose of 6 mg/kg (3–6 yo) | 2057 | N/A | N/A | 1109 | N/A | N/A |
| Simulated dose of 6 mg/kg (6–12 yo) | 1412 | N/A | N/A | 795 | N/A | N/A |

The studied pediatric groups were divided into two age ranges: 3–5 y/o and 6–12 y/o. Using an age group of 3–5 rather than 3–6 y/o was due to the age setup in Simcyp® that does not allow for age overlap.

Since the published clinical data have shown a higher exposure of the drug in pediatrics after administration of the approved sofosbuvir dose, various weight-based doses were explored to predict a dosing regimen for pediatric patients. The profile of the tested dose regimen was compared to the FDA-approved doses of 150 mg and 200 mg.

The generated pharmacokinetics values were further compared with three different pediatric published data, Table 4.

**Table 4.** Comparison of physiologically based pharmacokinetic (PBPK) model prediction and clinically observed values for PK parameters in healthy adult population for 400 mg oral sofosbuvir dose.

| Published Studies | AUC (µg·h/L) | | | Cmax (µg/L) | | |
|---|---|---|---|---|---|---|
| | Pred | Obs | Fold (Pred/Obs) | Pred | Obs | Fold (Pred/Obs) |
| * Hill et al. [21] | 2051 | 1123 | 1.8 | 545 | 531 | 1.02 |
| ** Abdallah et al. [23] | 2051 | 1105 | 1.9 | 545 | 910 | 0.6 |
| ** Majnooni et al. [24] | 2051 | 1057 | 1.9 | 545 | 387 | 1.4 |
| ** German et al. [29] | 2051 | 1370 | 1.5 | 545 | 703 | 0.8 |

Used in: * model development, ** model verification.

## 3. Results

### 3.1. Model Development

The sofosbuvir model was successfully built in the virtual healthy adult population in Simcyp® V20. The simulated outputs were comparable to the published pharmacokinetic

parameters [21]. Simulation of plasma concentrations of 400 mg sofosbuvir in healthy adults generated pharmacokinetics predictions of AUC = 2051 µg·h/L and $C_{max}$ = 545 µg/L. All folds were within the accepted range of 0.5–2. The PK parameters and folds are summarized in Table 4 and Figure 2.

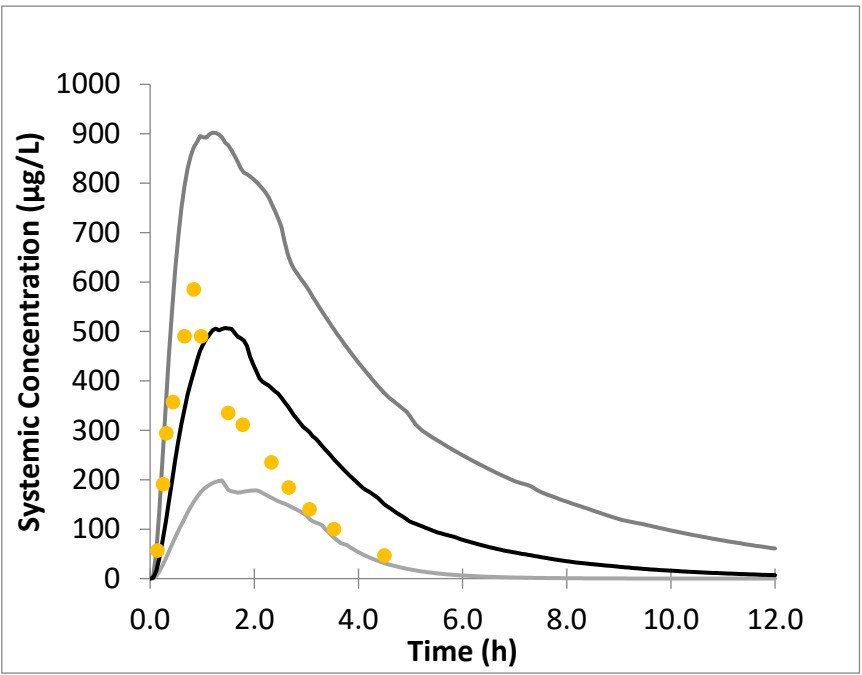

**Figure 2.** Simulated plasma concentration–time profile following oral administration of 400 mg sofosbuvir in healthy adult population. The gray lines correspond to the 95% and 5% margins of the data, the black line is the mean simulated data, and the yellow dots represent the observed adult data [21].

The data were presented as a single rather than multiple dose simulation as no accumulations of the drug or metabolites were expected [26]. Two clinical studies have confirmed the superposition of sofosbuvir PK upon multiple dosing suggesting no enzyme inhibition or induction [20,28].

### 3.2. Model Verification

Subsequent to the sofosbuvir model development, a model validation was performed using three additional published clinical studies that were not used in the compound model development. The additional studies are listed in Table 4. The AUC folds of (pred/obs) were consistent with all studies and within the range of 0.5–2. There is a $C_{max}$ variability between different clinical studies, but all folds were within the accepted range of 0.5–2. In addition, a high $t_{max}$ variability was reported. The $t_{max}$ can vary from 0.5 h to 2 h, representing 4 fold of intersubject variability; thus, it was excluded in model development and verification [23].

### 3.3. Paediatric Simulation

The PK simulated parameters of the approved 150 mg daily dose for 3–6 y/o age range patients were compared to a pediatric sofosbuvir clinical study performed on the same age group [30].

However, the 3–6 y/o results reported by Rosenthal et al. were excluded since the authors reported the sofosbuvir PK of one patient only [31].

The model successfully predicted the PK of sofosbuvir in this age range as the fold errors for AUC and $C_{max}$ (predicted/observed) were within the accepted range of 0.5–2.

Similarly, the PK simulated AUC and $C_{max}$ of the approved 200 mg daily dose for 6–12 y/o age range patients were within the accepted range of 0.5–2 compared to the observed data [31,32], thus confirming that the developed sofosbuvir model can be effectively used to evaluate a pediatric dose (Figure 3, Table 4).

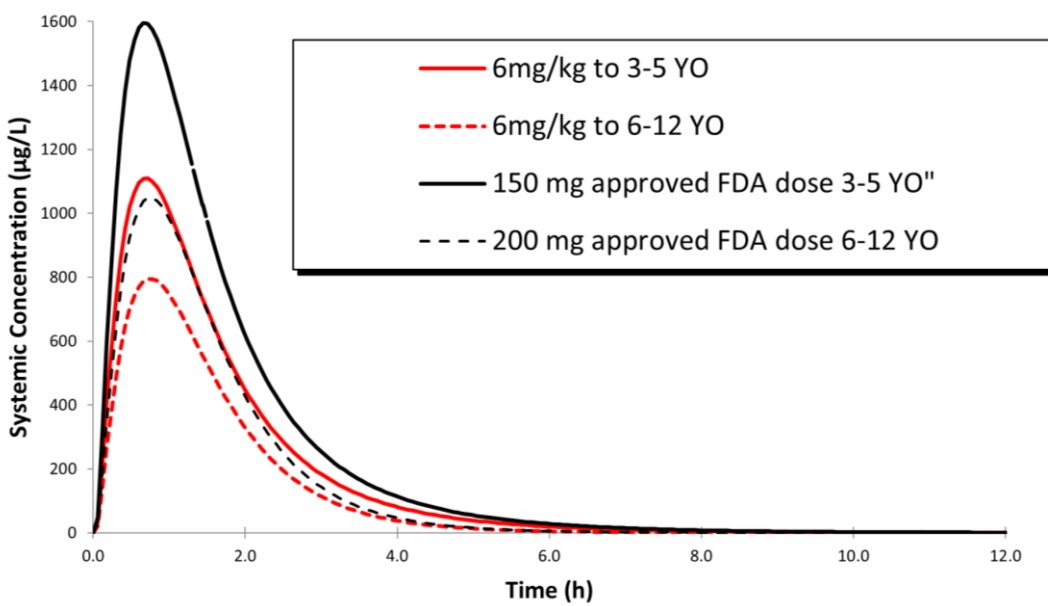

**Figure 3.** Simulated plasma concentration–time profile following a dose of 200 mg sofosbuvir to children ages 6–12 y/o, a dose of 150 mg sofosbuvir in children ages 35 y/o and a dose of 6 mg/kg sofosbuvir to children ages 3–5 y/o and 6–12 y/o.

Using this model, to simulate a weight-based dose of sofosbuvir to pediatric patients, a dose of 6 mg/ kg was found to provide an equitable exposure of the drug to adults (Figure 3, Table 5).

**Table 5.** Comparison of PK parameters of clinically observed sofosbuvir and GS-331007 upon administration of approved doses of 150 mg and 200 mg in Pediatric population and 400 mg in adults.

| Observed | HCV Adult Patients 400 mg [7] | Pediatric Patients (3–6 y/o) 150 mg [30] | | * Pediatric Patients (6–12 y/o) 200 mg [31,32] | |
|---|---|---|---|---|---|
| Sofosbuvir | | | % ped/adult | | % ped/adult |
| AUC (µg·h/L) | 1030 | 2500 | 243% | 1280 | 124% |
| $C_{max}$ (µg/L) | 511 | 1420 | 278% | 738 | 144% |
| Active Metabolite GS-331007 | | | | | |
| AUC (µg·h/L) | 7120 | 11,700 | 164% | 7895 | 110% |
| $C_{max}$ (µg/L) | 582 | 1000 | 172% | 839 | 144% |

* Presented parameters are an average of two clinical studies.

The 6 mg/kg dose provided an approximate 70–77% lower exposure of sofosbuvir compared to the fixed 150 and 200 mg doses (Table 6).

**Table 6.** The PK parameters of predicted sofosbuvir upon administration of approved doses of 150 mg and 200 mg compared to the simulated dose of 6 mg/kg in pediatric population.

| Pediatric Age Range | (3–6 y/o) | | | (6–12 y/o) | | |
|---|---|---|---|---|---|---|
| Predicted | 150 mg | 6 mg/kg | % Predicted 6 mg/kg/150 mg | 200 mg | 6 mg/kg | % Predicted 6 mg/kg/200 mg |
| AUC (µg·h/L) | 2908 | 2057 | 71% | 1842 | 1412 | 77% |
| Cmax (µg/L) | 1586 | 1109 | 70% | 1048 | 795 | 76% |

## 4. Discussion

The introduction of directly acting antivirals has revolutionized the treatment of HCV globally. Using the optimal dose of such antivirals in different populations is critical in eradicating the virus. Sofosbuvir is an effective therapy option in the treatment of HCV; when used in combination with other antivirals such as daclatasvir, ledipasvir, or velpatasvir, as asustained virologic response has been shown. The FDA approved the use of this antiviral in adults and recently in pediatrics.

The pediatric dose is frequently overpredicted when extrapolated from adults' dose [33]

Simcyp® enables the simulation of the PK changes in different pediatric age ranges by adjusting the input physiological parameters.

Our sofosbuvir compound model was successfully developed and verified using Simcyp® in the healthy adult population, and then used to predict and evaluate the optimal pediatric dose.

Owing to the weight variation in pediatrics, providing a fixed dose can possibly result in an under- or overexposure of the drug; however, to minimize exposure variability, a weight-based dosing will be superior to the fixed-dose regimen. The estimated pediatric dose that exposes the patient to a similar $C_{max}$ and AUC of an adult was found to be 6 mg/kg.

Upon reviewing the published clinical data of sofosbuvir and its active metabolite in patients, it is evident that the approved pediatric dose provides a reasonable exposure in the 6–12 y/o age range (110–144%) compared to adult patients (Table 5). However, in the age group 3–6 y/o, it yielded a much higher exposure. The AUC of the drug and its metabolites increased by 243% and 164%, respectively, and the $C_{max}$ of the drug and its metabolites increased by 278% and 172%, respectively. This justifies the need for adjusting the dose for the 3–6 y/o age range. An exposure reduction of approximately 30–35% will reduce the AUC and $C_{max}$ values to provide a reasonable exposure. Our simulation shows that a weight-based dose of 6 mg/kg provides an estimated 29–30% exposure reduction compared to the current approved dose (Table 6).

The renal ontology of the pediatric population continues to develop until adolescence. In other words, the kidney function of a 3 y/o is significantly lower than an adult. A significant change in the glomerular filtration rate (GFR) happens during the first year and continues until adolescence. This should be considered in dosing medications with primary renal elimination, which is the case of the active metabolite of our drug.

It is worth mentioning that all studies excluded liver cirrhosis patients, and according to one study, there is an increase in AUC and $C_{max}$ of sofosbuvir in patients with compromised liver function compared to that of patients who have normal liver function.

Thus, the dose adjustment for this particular population should be taken into consideration to ensure active metabolite GS-331007 optimal exposure, especially since many HCV patients do suffer from compromised liver function [25].

Another population that may require special attention in sofosbuvir dosing is patients with low renal function. Chronic HCV is associated with a risk of renal deterioration over time. The drug's active metabolite is excreted renally; thus, compromised renal functions necessitate dose adjustment.

In comparing the PK of healthy volunteers and HCV-infected patients, it was noted that the AUC and $C_{max}$ of sofosbuvir were higher in infected patients, but lower for the active metabolite (GS-331007) [7].

In our study, we have used the healthy volunteers' data to develop the compound model to minimize the effect of the disease on the drug PK. In addition, the number of subjects enrolled in the healthy adult studies was higher than the HCV patients' studies. However, in the final analysis we used the infected adult data as a point of comparison to the pediatric patients.

## 5. Conclusions

A new PBPK sofosbuvir compound model was successfully developed to simulate drug PK in pediatrics. This Simcyp® model can be a useful tool in simulating the drug's performance in different populations with varying demographics, and to support relevant decision making in clinical settings. Using the developed model, the study provides strong evidence that the defined sofosbuvir weight-based dosing regimen of 6 mg/kg provides an appropriate drug exposure in pediatric patients ages 3–6 y/o. However, no dosage adjustment is deemed necessary in children ages 6–12 y/o. Further clinical study is necessary to adjust the dose.

**Author Contributions:** Conceptualization, R.E. and E.A.; methodology, R.E., E.A. and A.A.; software, R.E., E.A. and A.A.; validation, R.E. and E.A.; formal analysis, R.E. and E.A.; investigation, R.E. and E.A.; resources, E.A.; data curation, A.A., A.S.G., A.M.A.-M. and O.A.; writing original draft preparation, R.E., E.A. and A.A.; writing—review and editing, R.E., E.A., A.S.G., A.M.A.-M. and O.A.; visualization, R.E. and E.A.; supervision, E.A.; project administration, E.A.; funding acquisition, NA. All authors have read and agreed to the published version of the manuscript.

**Funding:** No external funding was received to perform this research, It was performed under the employer (West Coast University) of the first and corresponding authors in West Coast University.

**Institutional Review Board Statement:** Not applicable.

**Informed Consent Statement:** Not applicable.

**Data Availability Statement:** The data presented in this study are openly available in different papers refered to in Tables 2–5 and cited in the reference section. There is no newly generated data, our paper is based on an analysis of published data and we have cited all of them.

**Acknowledgments:** The authors would like to acknowledge Certara for providing the complementary Simcyp® license and technical support.

**Conflicts of Interest:** The authors declare no conflict of interest.

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
