# Peer review of "PBPK Evaluation of Sofosbuvir Dose in Pediatrics Using Simcyp®"

_scipharm, doi:10.3390/scipharm91030038_

Round 1

Reviewer 1 Report

The current manuscript is an interesting study on the determination of the pharmacokinetics of a specific drug, sofosbuvir, using a computational tool, Simcyp®. The methodology and results are well described. Nevertheless, a few small changes should be made before acceptance for publication:

- Since the authors used a computational tool for pharmacokinetic modeling and simulation, a discussion of the relevance of this type of tools should be made in the introduction section, with mention of the great advantage that in silico tools can have for both science and healthcare;

- The text should be more fluid, with less “very small” paragraphs;

- A more thorough description of the drug sofosbuvir should be done, namely its advantages compared to other drugs for the same therapeutic purpose;

- Figure 1 quality should be improved (resolution);

- All Tables and Figures should be mentioned in the text, and then appear immediately after they are first mentioned;

- How does Simcyp compare to other tools for optimum dose determination, especially other in silico tools? This should be addressed.

Author Response

The current manuscript is an interesting study on the determination of the pharmacokinetics of a specific drug, sofosbuvir, using a computational tool, Simcyp®. The methodology and results are well described. Nevertheless, a few small changes should be made before acceptance for publication:

- Since the authors used a computational tool for pharmacokinetic modeling and simulation, a discussion of the relevance of this type of tools should be made in the introduction section, with mention of the great advantage that in silico tools can have for both science and healthcare;

            This paragraph has been added

Motivated by the universal acceptance that in silico calculations, and in vitro data, can provide insight into pharmacokinetic processes and reduces the time, effort, and money spent in the drug development process, many pharmaceutical and biotechnology companies heavily applied modeling and simulation in pharmacokinetics.

An in silico user-friendly tool allows flexibility to design models of concepts that go beyond the software capabilities, which allows for the broader utilization of the software for many scientists who have no programming proficiencies 11. A few examples of simulation software programs are Simcypâ Gastro PlusTM, GI-Sim ,and PK Sim are used for this purpose. Some PK research groups prefer to develop their own in-house models should they have the needed coding skills.

Few studies aimed to explore the strengths and restrictions of the simulation software programs. The average performance was relatively consistent across many software platforms. And the accuracy of the prediction is dependent on the type of drug.

Some programs have better prediction for drugs with challenging solubility and incomplete absorption while others create more accurate models for drugs with challenging elimination and metabolism or models involving drug- drug interactions. Some studies suggest that modeling and simulation groups should perform model evaluations using multiple software to decide on the best model, but the reality is that there is a major cost hindrance to this ideal practice.

- The text should be more fluid, with less “very small” paragraphs;

The text is made more fluid, small paragraphs combined

- A more thorough description of the drug sofosbuvir should be done, namely its advantages compared to other drugs for the same therapeutic purpose;

The following information has been added regarding the directly acting antivirals (DAA), as a class of drugs. There is no clear advantages of one over the other, they just have different mechanisms. They are used in combinations to provident the development of resistant viruses. The mechanism of action of Sofosbuvir was already in described in the text

Before the discovery of directly acting antivirals (DAA), treatment of chronic HCV has been based completely on a combination of pegylated interferon and ribavirin. Fortunately, with the discovery of the DAA in 2014, chronic HCV infection became curable to almost 100 %. The DAA regimens resulted in a paradigm shift in the handling of the disease. Its an example of victory in the fight against infections. It allowed for the complete global eradication of the virus 5 Sofosbuvir, is a DAA of distinct interest due to its effectiveness, low side effects, oral administration, and high barrier to resistance

The high rates of HCV replication and possible mutation permit HCV to rapidly mutate. Using the combination DAA therapy in targeting different viral functions and stages of the virus life cycle is an approach to limit resistance. A similar tactic is applied when using cocktail anti-HIV to avoid resistance to monotherapy with the difference that the DAA therapy is administered for a limited time, unlike the HIV which is a lifetime therapy. 5

- Figure 1 quality should be improved (resolution);

A new file is uploaded

- All Tables and Figures should be mentioned in the text, and then appear immediately after they are first mentioned;

Done

- How does Simcyp compare to other tools for optimum dose determination, especially other in silico tools? This should be addressed.

A few examples of simulation software programs are Simcypâ Gastro PlusTM, GI-Sim ,and PK Sim are used for this purpose. Some PK research groups prefer to develop their own in-house models should they have the needed coding skills.

Few studies aimed to explore the strengths and restrictions of the simulation software programs. The average performance was relatively consistent across many software platforms. And the accuracy of the prediction is dependent on the type of drug.

Some programs have better prediction for drugs with challenging solubility and incomplete absorption while others create more accurate models for drugs with challenging elimination and metabolism or models involving drug- drug interactions. Some studies suggest that modeling and simulation groups should perform model evaluations using multiple software to decide on the best model, but the reality is that there is a major cost hindrance to this ideal practice.

Reviewer 2 Report

The authors of the manuscript used Simcyp® software (Version 20, Simcyp Division, Certara UK Limited, United Kingdom) to investigate the effect of pharmacokinetic variability in two pediatric age groups for sofosbuvir. The physicochemical properties of the drug and clinical data from a healthy adult population were used to develop the model. The objective of the manuscript is clear and the figures and tables are informative.

Some observations:

- In Table 5, commas are used instead of dots, and columns may have slipped in the pagination. 

- Reference 28 is not appropriate in this form

- Why is 12 years the upper limit for the second age group for children?

- Sofosbuvir is included in combination therapy in most regimens. For example, it is used in combination with ribavirin or ledipasvir. Does this have an impact on the data predicted in the manuscript? Is it possible to look at drug combinations with Simcyp software or do you have data on this?

Author Response

The authors of the manuscript used Simcyp® software (Version 20, Simcyp Division, Certara UK Limited, United Kingdom) to investigate the effect of pharmacokinetic variability in two pediatric age groups for sofosbuvir. The physicochemical properties of the drug and clinical data from a healthy adult population were used to develop the model. The objective of the manuscript is clear and the figures and tables are informative.

Some observations:

- In Table 5, commas are used instead of dots, and columns may have slipped in the pagination. 

The commas have been removed, so all tables are consistent

- Reference 28 is not appropriate in this form

Fixed

- Why is 12 years the upper limit for the second age group for children?

As the adult dose is for 13 and up

- Sofosbuvir is included in combination therapy in most regimens. For example, it is used in combination with ribavirin or ledipasvir. Does this have an impact on the data predicted in the manuscript? Is it possible to look at drug combinations with Simcyp software or do you have data on this?

Even though that DAA can ave some DDI with other classes, There is no reported DDI between DAA

(Reference) Dick TB, Lindberg LS, Ramirez DD, Charlton MR. A clinician's guide to drug-drug interactions with direct-acting antiviral agents for the treatment of hepatitis C viral infection. Hepatology. 2016 Feb;63(2):634-43. doi: 10.1002/hep.27920. Epub 2015 Sep 21. PMID: 26033675.

A paragraph has been added about :

Sofosbuvir is always given in combinations as a technique to avoid resistance (like HIV).